# GazeSAM: Interactive Image Segmentation with Eye Gaze and Segment Anything Model

**Bin Wang**                                                    BIN.WANG@NORTHWESTERN.EDU
**Armstrong Aboah**
**Zheyuan Zhang**
**Hongyi Pan**
**Ulas Bagci**
*Northwestern University, Chicago, IL, 60611, USA*

## Abstract

Interactive image segmentation aims to assist users in efficiently generating high-quality data annotations through user-friendly interactions such as clicking, scribbling, and bounding boxes. However, mouse-based interaction methods can induce user fatigue during large-scale dataset annotation and are not entirely suitable for some domains, such as radiology. This study introduces eye gaze as a novel interactive prompt for image segmentation, different than previous model-based applications. Specifically, leveraging the real-time interactive prompting feature of the recently proposed Segment Anything Model (SAM), we present the *GazeSAM* system to enable users to collect target segmentation masks by simply looking at the region of interest. GazeSAM tracks users' eye gaze and utilizes it as the input prompt for SAM, generating target segmentation masks in real time. To our best knowledge, GazeSAM is the first work to combine eye gaze and SAM for interactive image segmentation. Experimental results demonstrate that GazeSAM can improve nearly 50% efficiency in 2D natural image and 3D medical image segmentation tasks. The code is available in https://github.com/ukaukaaaa/GazeSAM.

**Keywords:** Eye Gaze, Segment Anything Model, Interactive Image Segmentation, Eye Tracking

## 1. Introduction

In the field of modern machine learning and computer vision, having large amounts of data and accurate labels is crucial for generalizable deep learning models with high accuracy. Accurate labeling is particularly important to ensure these models work effectively. However, labeling data can be extremely time-consuming and labor-intensive, especially for the image segmentation task. For some fields, such as radiology, this should be done with care as it is a high-risk task and often associated with diagnostic or prognostic decisions. Traditionally, labeling involves people manually drawing lines at the boundaries of the objects or regions of interest. To expedite and improve the efficiency of this process, interactive image segmentation was introduced, combining human interaction input and task-specific automated models.

For the human interaction part, there are several existing types, such as clicks (Chen et al., 2021; Lin et al., 2020; Sofiiuk et al., 2020, 2022; Liu et al., 2022; Chen et al., 2022), scribbles (Bai and Wu, 2014; Grady, 2006; Li et al., 2004), bounding boxes (Lempitsky et al., 2009; Rother et al., 2004; Wu et al., 2014), and polygons (Acuna et al., 2018). However,

these methods are all based on mouse interaction. When facing the workload of large-scale datasets with high dimensional nature and low-resolution context, such as in radiology scans, the annotator will easily get tired after repeatedly clicking for precise annotations. This can largely decrease the efficiency of the data labeling process and the quality of the data annotations. To solve this problem, one promising approach that has not yet been explored extensively is the use of eye gaze to perform interactive image segmentation. While previous research regarding eye gaze has primarily focused on understanding the relationship between human attention and cognitive decision-making (Wood et al., 2020; Khosravan et al., 2019), its potential in automating the interactive segmentation has yet to be fully realized.

By tracking annotators' eye movements when they label the images, it is possible to identify the regions of interest that are most relevant to them using their gaze points. This rich information can be leveraged to segment the images automatically, and there are multiple advantages to using eye gaze as the interaction input. **First,** it is more natural and intuitive because eye gaze-based interaction aligns with how humans naturally perceive objects by simply looking at them. If we take the eye gaze as the prompt for segmentation, it would be more intuitive and user-friendly. **Second,** it can reduce user fatigue significantly. Using a mouse to annotate large-scale datasets will lead to a tedious click job. Interacting by eye gaze is way easier and less fatiguing since users do not need to perform physical movements to mark the object or draw the bounding boxes. **Third,** eye gaze enables faster and more efficient interactions. Users can simply glance at the object they want to segment without the need for mouse clicking or drawing. **Fourth,** using eye gaze as interaction input can generate multiple prompt input points in less than one second, which can input more information into the automated model in a short time. This will increase the accuracy of the generated segmentation masks. But, mouse-based interaction can only allow annotators to click to generate prompt input points one by one individually.

Related to the automated model part, previous studies (Chen et al., 2022; Liu et al., 2022; Sofiiuk et al., 2022) have trained separate models for different segmentation tasks for vision and radiology applications. This leads to redundant model training efforts and requires the annotators to switch between models for annotations, which is unnecessary. One solution is to employ the Segment Anything Model (SAM) (Kirillov et al., 2023) instead of separate models. While SAM has shown tremendous success across various domain applications (Ma and Wang, 2023; Wang et al., 2023; Cen et al., 2023), its potential to form part of an interactive image segmentation system has not been entirely studied yet. Thanks to SAM's ability to generalize to new domains or even unseen objects and its promptable structure, we could build a general interactive image segmentation system instead of training separate models for different tasks. More importantly, SAM can enable the system to conduct seamless, real-time interactive segmentation because of its lightweight architecture.

Hence, we propose the *GazeSAM* to investigate the feasibility and efficacy of integrating eye gaze with SAM for an interactive image segmentation system. The proposed system uses eye-tracking technology to identify the regions that annotators are interested in and then prompts those attention cues to the SAM model to segment the images accordingly. The system is designed to be user-friendly, accurate, and fast in generating segmentation results.

The major contributions of this work are summarized as follows:

1. We propose a novel interactive image segmentation system, called *GazeSAM*, that combines eye gaze with SAM for an efficient and user-friendly data annotation process.

2. Instead of the mainstream mouse-based interaction methods, we introduce an alternative interaction type, *eye gaze*, which is more natural, intuitive, efficient, and less fatiguing.

3. *GazeSAM* utilizes SAM's zero-shot power and lightweight architecture to avoid training separate task-specific models and enable real-time interaction.

4. Our system has the unique capability of operating with 2D and 3D images, typically used in medical settings. This is the first of its kind developed to significantly increase radiologists' work efficiency in daily clinical practice.

5. We have evaluated *GazeSAM* on 2D image segmentation datasets GrabCut (Rother et al., 2004) and Berkeley (Martin et al., 2001), 3D medical image segmentation dataset (Bilic et al., 2023). Our results show a significant efficiency improvement without deviation from the high accuracy.

## 2. Related Work

Interactive image segmentation has always been a popular topic in the computer vision field. The reason is that the training of the models requires large amounts of data and high-quality annotations. Traditional annotation methods involve manually marking the object boundary, which is labor-intensive and tedious. DIOS (Xu et al., 2016) incorporates the automated models into the interactive image segmentation task, which increases the efficiency of the labeling significantly. The users are able to segment the target regions by conducting clicks to input the positive and negative prompts for the automated model. Then, the model generates the segmentation mask according to the users' prompt.

After this, SimpleClick (Liu et al., 2022), FocalClick (Chen et al., 2022), BRS (Jang and Kim, 2019), and f-BRS (Sofiiuk et al., 2020) are proposed to conduct interactive image segmentation based on the mouse clicks. Lempitsky et al. (Lempitsky et al., 2009) utilize the bounding box to replace the single mouse click. Bai et al. (Bai and Wu, 2014) introduce scribble as the interactive type, in which they ask the user to roughly draw lines on the target object. Acuna et al. (Acuna et al., 2018) propose a polygon interactive method to enable the users to mark the object boundary during annotation. However, these methods are all mouse-based interaction methods, which leads to user fatigue when doing large-scale dataset annotations. In this paper, we introduce eye gaze-based interaction, which is more natural and user-friendly.

## 3. Methods

In this section, we describe our proposed framework *GazeSAM* for real-time segmentation mask collection by utilizing a screen-based eye-tracker and Segment-Anything Model (SAM). As illustrated in Fig. 1, *GazeSAM* comprises two parts: eye-gaze data collection and segmentation.

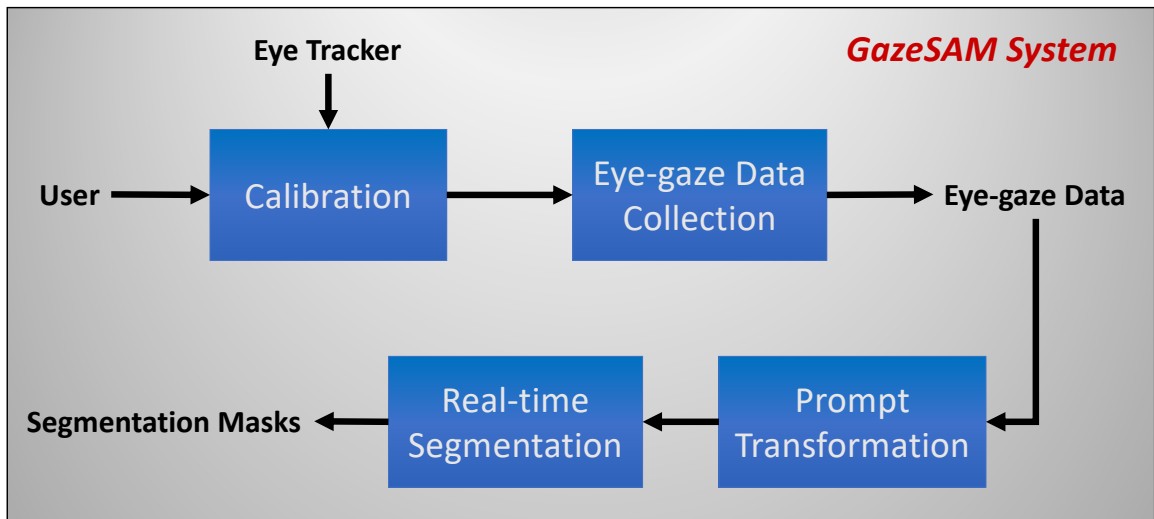

Figure 1: Overview of our proposed system.

### 3.1. Eye-Gaze Data Collection

In this study, a Tobii Pro Nano screen-based eye tracker is used. It is a small, lightweight, and easy-to-use eye tracker whose length is 170mm, weight is 59g, and the sampling rate is 60Hz.

Before the experiment, calibration of the eye tracker is required because it ensures the eye movement is tracked accurately and makes the gaze coordinate on the screen consistent with where the user is looking. Here, we adopt a five-point calibration procedure in Tobii Pro eye tracker manager. After completing the calibration, the eye-gaze data can be collected in the form of the location coordinate on the screen.

### 3.2. Prompt Transformation & Segmentation

The prompt encoder in SAM is designed to support a wide range of prompt formats, such as points, boxes, and text. To integrate eye-gaze data as a new type of prompt into the SAM, we need to first conduct a prompt transformation.

Eye-gaze data can be considered as a sequence of scatter points that correspond to the eye movement over time. Hence, it is possible to transfer the eye-gaze data into a point or a sequence of points, which can be utilized as the point prompt for SAM. Prior to this, it is necessary to first solve the coordinate problem. The eye-gaze points coordinates, denoted as $S_1, S_2, ...S_n$ are collected in the screen coordinate space. We need to transform it into the image coordinate space as follows:

$$I_1, I_2, ..., I_n = f(S_1, S_2, ...S_n), \tag{1}$$

where $f(\cdot)$ is the mapping function between two coordinate space and $I_1, I_2, ..., I_n$ are the eye-gaze points coordinates in image coordinate space.

Then, as illustrated in Fig. 2, *GazeSAM* supports two options for inputting the eye-gaze data as a prompt for SAM. The first option is to use the whole sequence of eye-gaze points

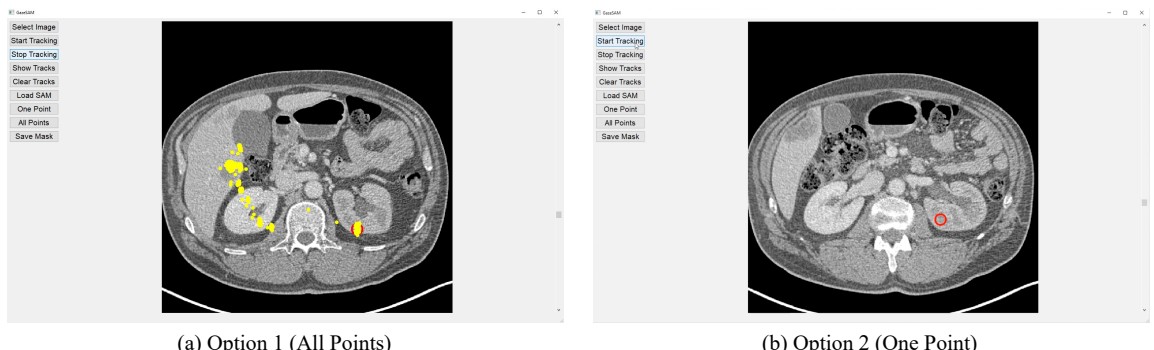

(a) Option 1 (All Points)  (b) Option 2 (One Point)

Figure 2: Two eye-gaze prompt options for segmentation in *GazeSAM*.

collected over time, which can provide a more comprehensive representation of the user's gaze trajectory. The second option is to use the eye-gaze point collected at the last time point as the prompt. This approach is more appropriate when a coarse segmentation mask of a single object is desired.

It is noted that SAM might not always generate a perfect segmentation mask, especially for the boundary regions. To refine the generated mask, users need to manually add points to those regions, which can be tedious and time-consuming. In the first option, *GazeSAM* simplifies this process by allowing users to add points by simply looking at the desired areas. In this way, a more efficient approach to refine the segmentation mask is offered, which has the potential to greatly enhance the user experience and speed of the whole pipeline.

Given a pre-computed image embedding and the prompt transformed from eye-gaze data, SAM can generate a segmentation mask subsequently in near real-time, making it an interactive segmentation system by using eye-tracking technology.

## 4. Experiments

### 4.1. Interactive Image Segmentation User Interface

As illustrated in Fig. 3(a), we develop a user interface for the *GazeSAM* system to conduct the interactive image segmentation. It incorporates multiple functions for the users. The function panel is situated on the left side of the interface, and each function can be easily activated either using corresponding keyboard shortcuts or clicking the buttons. "Select Image" is for choosing the 2D or 3D image files to annotate. In the following experiments, we enable the user to choose a folder that contains all the image files and switch the next and last image on the screen by typing the left or right key. Once the user clicks the "Start Tracking" button, the eye-gaze point is displayed as a hollow red circle, which tracks the user's eye movement in real time. The red circle will follow the trajectory of the user's eye as it moves across the screen. When the experiment is finished, "Stop Tracking" helps close the eye tracking system. The "Show Tracks" option displays the eye movement trajectory. These eye movement tracks are composed of multiple eye-gaze points, as yellow dots in Fig. 2(a), that can be taken as the model prompt input. "Clear Tracks" enables the user to delete current eye-gaze points and restart the eye gaze recording. For

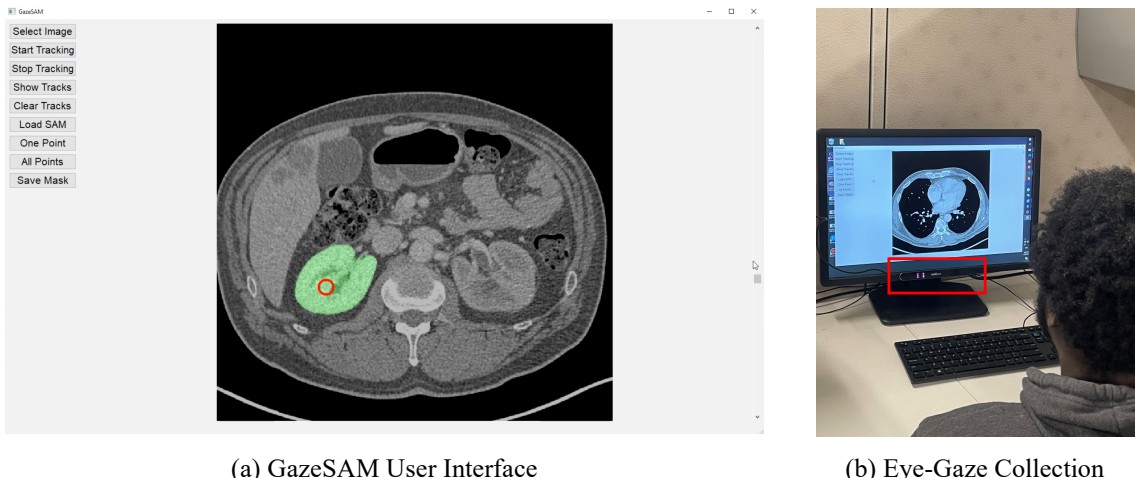

(a) GazeSAM User Interface          (b) Eye-Gaze Collection

Figure 3: Our designed user interface and experiment setting of eye-gaze data collection.

convenience, we also allow the user to press key "A" to clear the eye gaze and restart automated segmentation. "Load SAM" reads SAM pretrained weight at the beginning of the experiments. For segmentation, GazeSAM provides two options ("One Point" and "All Points") as described in Section 3.2. Once an option is activated, a segmentation mask is generated in real-time and shown in green color. This mask is automatically updated based on the location of the user's eye gaze, allowing for dynamic adjustments to the segment target or iterative refinement of the segmentation. Users can save the eye-captured segmentation mask displayed on the interface by using the "Save Mask" function at any point during the process. For 3D medical images, a scroll bar is provided on the right side of the interface to control the slice image selection.

## 4.2. Experiment Settings

As illustrated in Fig. 3(b), the eye tracker is positioned directly below the lower edge of the display, and the user maintains a viewing distance of approximately 60cm from the screen.

During the experiment, the annotator first loads SAM pretrained weight and then conducts the calibration for the eye tracker. After that, the test image list is read, and we start to record the annotation time from the first image and end the time after the last image labeling.

Due to the absence of mouse clicks in eye gaze-based interaction, we can not employ the primary evaluation metric, Number of Clicks (NoC) at Intersection over Union (IoU), in our comparative experiment. Instead, we assess efficiency improvement by directly comparing annotation time costs. Additionally, we utilize the mean Intersection over Union (mIoU) as the metric to evaluate accuracy.

To ensure a fair comparison, we exclusively select SAM as the backbone model and assess efficiency by comparing eye gaze-based interaction with mouse-based interaction.

### 4.3. 2D Interactive Image Segmentation

In this experiment, the annotator is asked to label the data on the GrabCut (Rother et al., 2004) and Berkeley (Martin et al., 2001) dataset. The GrabCut dataset comprises 50 images, each containing a single object, while the Berkeley dataset consists of a total of 100 images.

Table 1: Quantitative results on GrabCut and Berkeley dataset.

| Model | GrabCut | | Berkeley | |
| --- | --- | --- | --- | --- |
| | Time/s ↓ | mIoU/% ↑ | Time/s ↓ | mIoU/% ↑ |
| GazeSAM | **125** | 92.10 | **266** | 85.56 |
| SAM + Mouse | 232 | 92.31 | 424 | 88.33 |

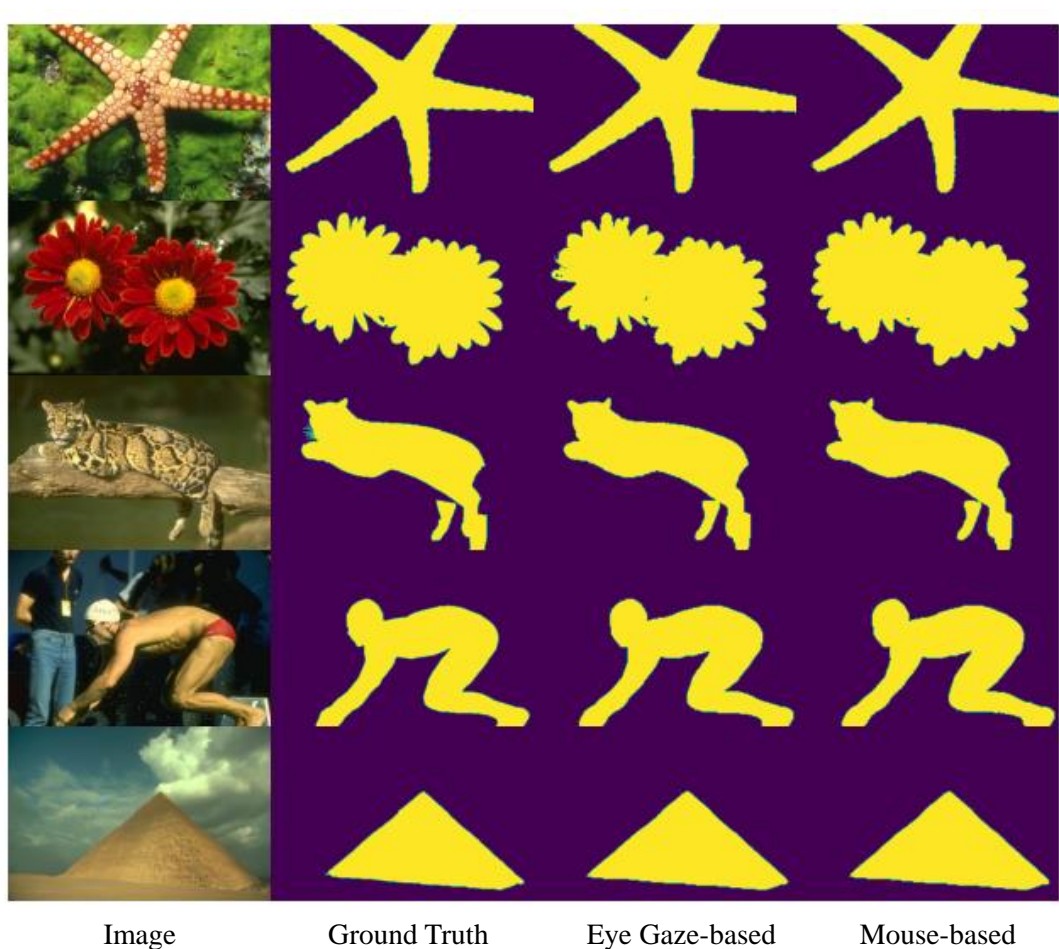

Image   Ground Truth   Eye Gaze-based   Mouse-based

Figure 4: Examples from GrabCut and Berkeley dataset.

The results presented in Table. 1 demonstrate that annotators can complete segmentation labeling work in nearly 50% less time when using the eye gaze-based interaction system

compared to the mouse-based interaction system. This highlights the efficiency advantages of the eye gaze-based approach.

Moreover, when we compare the mIoU scores, it becomes apparent that the efficiency gains achieved by the GazeSAM system do not come at the expense of accuracy. The accuracy score is comparable between the two systems, demonstrating that eye gaze-based interaction offers an efficiency boost while maintaining accuracy comparable to the current mainstream mouse-based interaction system. From the visualization of generated segmentation masks in Fig. 4, we can also find that the accuracy performance between eye gaze-based and mouse-based interaction systems is similar.

The reason for the higher efficiency of eye gaze-based interaction is readily apparent. As illustrated in Fig. 5, users can rapidly obtain multiple input points simply by glancing at the target region in less than a second. In contrast, the mouse-based interaction requires the user to click the target area to refine the segmentation performance, which takes more than one second. Consequently, we can conclude that the eye gaze-based interaction system offers a distinct advantage over the mouse-based interaction system in terms of efficiency.

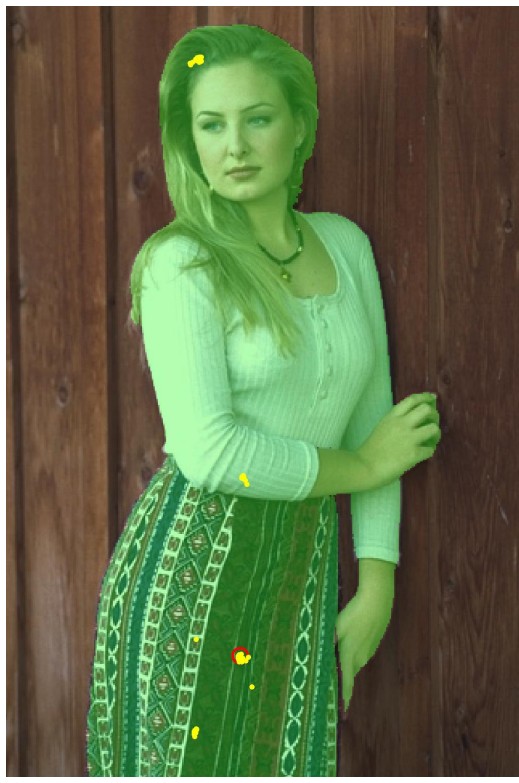
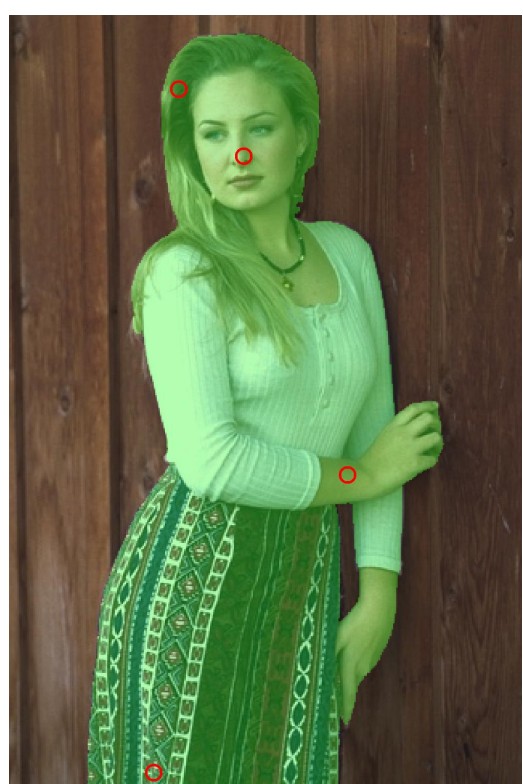

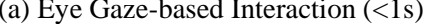

(a) Eye Gaze-based Interaction (<1s)     (b) Mouse-based Interaction (~1.5s)

Figure 5: Efficiency comparison between eye gaze-based interaction and mouse-based interaction.

### 4.4. 3D Interactive Medical Image Segmentation

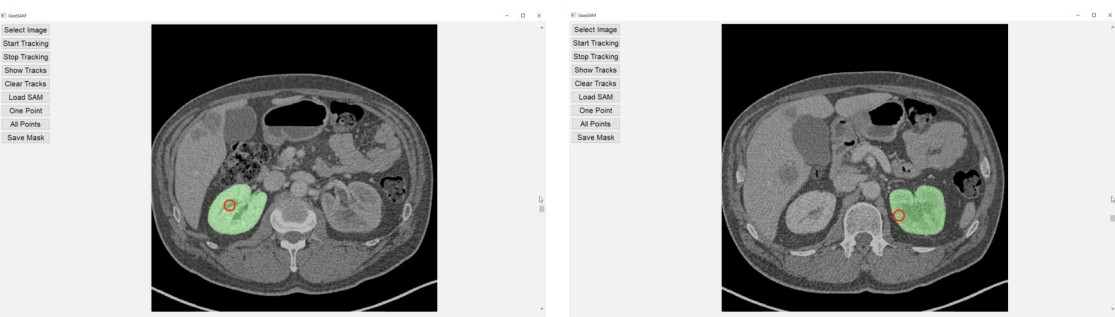

Figure 6: Some examples of 3D interactive medical image segmentation.

The GazeSAM system extends its capabilities to support 3D image segmentation, which we have successfully applied to the field of 3D medical image segmentation. This system is designed to be user-friendly for radiologists, enabling them to conveniently adjust the slice image and zoom in or out to gain a clearer view of the target area. As depicted in Fig. 6, we can observe that the organ has been successfully and clearly segmented. We have also provided a demo video for the 3D medical image segmentation in our GitHub repository. It is worth noting that the experience is exceptionally smooth when the radiologist stares at one specific organ and controls the slice using the mouse slicer. This leads to the potential for improving the efficiency of daily clinical workflows, offering radiologists a feasible tool for assisting precise and efficient medical image analysis.

Given that SAM is primarily trained on natural images, its ability to infer accurate segmentation on medical images is limited. While GazeSAM offers a more efficient approach to improve segmentation quality by simply looking at the desired areas and incorporating more eye-gaze prompts in regions with poor segmentation, its performance is still restricted in some cases. To overcome this limitation, fine-tuning SAM on a large-scale medical image dataset is a possible solution (Ma and Wang, 2023; Cheng et al., 2023). Besides, we can also use the generated mask as a coarse segmentation result for further refinement.

## 5. Conclusion

In this study, we propose GazeSAM, a novel interactive image segmentation system that utilizes eye gaze as the interactive prompt instead of the mouse-based interaction such as click, scribble, bounding box, and polygon. Our system takes advantage of SAM's zero-shot power and lightweight architecture to avoid training separate task-specific models and enable real-time interaction. By evaluating GazeSAM on 2D and 3D images, we observe that it offers significant efficiency improvement for the annotation workflow.

## Acknowledgments

This study is supported by NIH R01-CA246704, R01-CA240639, R15-EB030356, R03-EB032943, U01-DK127384-02S1, and U01-CA268808.

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
