# OpenReview forum: "GazeSAM: Interactive Image Segmentation with Eye Gaze and Segment Anything Model"
_NeurIPS.cc/2023/Workshop/Gaze_Meets_ML — Gaze Meets ML 2023 Poster_

### Official Review · Reviewer_PPPg · 2023-10-19
**Tool for 2D/3D image segmentation using eye gaze prompt**

**Rating:** 5
**Confidence:** 5

**Review:**

This paper proposes an interactive tool called GazeSAM for 2D/3D image segmentation using 2D eye gaze as input prompt. The paper combines the pre-trained Segment Anything Model (SAM) with the 2D eye gaze coordinate as a prompt to segment the region of interest in the image. Authors show that the GazeSAM achieves the same segmentation quality as a mouse click but in 50% less time. The paper is very well written, easy to understand, and provides qualitative evaluation. The authors claim to make the demo code public. To me, this is an interesting application of gaze tracking.

However, the authors claim that “GazeSAM is the first work to combine eye gaze and SAM for interactive image segmentation”, whereas the SAM authors show the use of eye gaze as a prompt to SAM for AR/VR applications [1].

Comments:
1. Did the authors evaluate “One point” vs. “All point” prompts for GazeSAM? It will be interesting to see how these two compare in time and mask quality.
2. Did the authors evaluate how the mask quality changes with the user-Tobii tracker calibration errors? This is important because users might need to move in between annotations and might need recalibration every time they move.
3. The qualitative examples of masks shown in the paper have larger RoI. Does the proposed tool work for segmenting smaller RoIs using eye gaze? This is important because, for segmenting smaller RoI, fixating at a particular point in an image is necessary, and microsaccades might be a problem. Also, it can cause fatigue to users after annotating a few images.
4. Just curious, is it possible to automate the click on the side panel also through gaze instead of mouse/keyboard clicking? It might reduce the time further.

Overall, the contribution is limited and needs more maturity before publication.

[1] https://ai.meta.com/blog/segment-anything-foundation-model-image-segmentation/

---

### Official Review · Reviewer_vJiW · 2023-10-23
**Simple evaluation of interactive segmentation tool with instant user feedback, combining SAM (Segment Anything) and eye-tracking data**

**Rating:** 6
**Confidence:** 3

**Review:**

The paper describes the development of an interactive 2D-segmentation tool using the zero shot capabilities of the Segment Anything foundational model in conjunction with eye-tracking data collected in real time. It shows that the tool reduces annotation speed by 50% while having just a small reduction in the quality of the results, when compared to using mouse to select points inside the thing to segment.

Strengths
- Interactive segmentation: the tool shows instantly what is included in the segmentation with each key point added, through both mouse and eye tracker. This show a strength of the employed pipeline: near real-time segmentation. It also makes the presented results in terms of time spent per segmentation and quality of segmentation be more representative of what a pipeline applied in the real-world is.
- Working tool: the authors say they will make the tool available on GitHub. The tool might be a good contribution for building segmentation datasets by people who have access to the same eye tracker.
- Reduction in time of annotation: the use of eye tracker for interactive segmentation does seem to reduce the time of annotation to around half of the time of using the mouse for the same task.

Weaknesses
- IRB or equivalent: the paper does not give enough details from the data collection sessions to indicate that it should be IRB exempt. It also does not specify the involvement of an IRB. The correct use of data collected from human subjects should be checked before the acceptance of the paper.
- Fatigue: the paper claims that the use of eye tracker instead of mouse helps with fatigue during data collection. The claim is included in the paper as something independent from the fact that the data collection takes half of the time with the eye tracker. However, it does not provide any reference or evidence to support that claim. I believe that those claims should be removed from the paper.
- Medical data without results: the paper does not give quantitative results for the medical data, so it would be hard for a potential user to evaluate if they should use this tool or not. Since the paper mentions restricted performance for medical data, results are probably of low quality. It would also be interesting to know how well the use of eye tracker does for thin organs, such as the pancreas. It is also worth noting that the SAM model performs slice-by-slice segmentation, and not volumetric segmentation, as some of the organs may require.
- Unclear impact of calibration: it is not clear what impact calibration has on the results. How much time does it take? What is the angular precision and inter/intra user variation? How often does it have to be repeated?
- Worse results with eye tracker: despite the claim that there is no impact on quality of segmentations when comparing the use of eye tracker against the mouse, there is a difference of 0.2 percentual points for one dataset and 2.8 percentual points for another dataset. The lack of confidence intervals for the results makes it hard to judge how significant that difference is.
- Lack of UX results: for a paper that contributes a tool, I would expect to have a deeper analysis of the user experience. The paper mentions the option to only use the last gaze location as a key point to add to the segmentation and the whole sequence of gaze data, but it does not compare them in any way. It would also be interesting to have user feedback in terms of what it is like to use the tool when compared to the use of a mouse.
- Single user experiments: since the experiments seem to have been run for a single user, it is not clear how much the results would generalize to new users.

I can’t verify the IRB worry as a reviewer without a discussion board with the authors. However, I believe that the paper provides a contribution in terms of showing what performance changes can be expected when using an eye tracker for interactive annotation of segmentations. The paper proposes to do a very simple analysis of the changes and seems to do them well for acceptance. More information could be gathered in a paper that does a deeper analysis.

---

### Official Review · Reviewer_4Ciq · 2023-10-24
**The authors propose an efficient, user-friendly, interactive and intuitive system for annotation of images using eye-tracking technology combined with Segment Anything Model (SAM).**

**Rating:** 7
**Confidence:** 4

**Review:**

Title: GazeSAM: Interactive Image Segmentation with Eye Gaze and Segment Anything Model

Purpose:
To develop an efficient, user-friendly, interactive and intuitive system for annotation of images using eye-tracking technology combined with Segment Anything Model (SAM).

Significance:
Uses eye-tracking technology to segment images instead of mouse clicks. Allows less tedious and more interactive system for generating annotations.

Method Proposed:
The authors have applied the Segment Anything Model (SAM) pre-trained on natural images. It is a promptable structure that can build a general interactive segmentation system. Some of the advantages of applying SAM is the light weight structure, uses zero-shot power, ability to generalize to new domains and unseen objects, ability to integrate with eye-gaze tracking.
The authors have used a small, lightweight eye tracker. The tracker is first calibrated to the user’s eye. Then a tracking of the eye gaze in the screen coordinate space is converted to the image coordinate space to generate the points which are fed to the system as points. They have two kinds, the first one track they eye movements from start to finish, other collects the last time point as prompt. If the segmentation needs more refining, more prompts can be added by looking at desired areas
Using these prompts, SAM generates segmentation masks in real time. The develop GUI allows to select the image, track eye movements in real time, generate prompts, use SAM to get segmentation results in real-time and finally save the masks. For 3D images, slices can be selected for annotations.

Strengths:
-	A new model that allows eye-gaze data to be used for segmentation of the images
-	This tool can be helpful especially for medical images, where usually the region of interests are small with irregular shapes and are too complex to manually segment using mouse
-	They can also be applied to improve accuracy - generating Regions of Interest (ROIs) are difficult using mouse clicks and convenient using eye-gaze
-	Time of annotations can be reduced

Weaknesses:
-	The results presented in Table 1 only provide two datasets that contain one object per image.
-	The accuracy of segmentation is comparable with mouse clicks but did not improve substantially
-	Authors did not provide information on the type of images used especially for 3D medical data
-	Authors can provide some results when multiple objects in the same image are segmented
-	No information of mIOU results for medical image segmentation on 3D data have been provided
-	No 2D medical image segmentation results have been provided
-	It is not clear if generalizability of the system can be established, since the datasets contain only one object per image
-	Information can be provided on the mapping function applied to convert the screen coordinate space data of the eye tracker to the image coordinate space.
-	In Figure 4, examples can be labeled to provide information on the image and its corresponding dataset

Recommendation:
-	Experiments and comparisons with more datasets may provide better information on the performance of the proposed approach

---

### Meta-Review · Area_Chair_MrwW · 2023-10-26

**Recommendation:** Accept (Poster)
**Confidence:** 4

**Metareview:**

This clearly written paper proposed an interactive gaze-based segmentation tool that harnesses SAM's point prompting features to segment unseen objects. Some baseline analyses comparing mouse versus gaze-based prompts were provided, including analysis of segmentation performance and speed differences. It would appear that only a limited subset of collected gaze was used in the system and important quantitative metrics, such as mIOU, are not shared for medical images, for which authors mentioned restricted performance. Given that utilizing gaze as prompts with SAM's flexible prompt input is something others have tried, the paper does not appear to have clear technical novelty. However, as a working gaze-based annotation tool available on GitHub, it might be a good contribution to other researchers in the field who have access to the same eye tracker. As an interactive GUI, the reviewers recommend more in-depth analysis of user experience and performance on different imaging domains to assist potential re-user with decision to use this tool or not. Overall, for the open source tool contribution, the recommendation is accept as poster presentation.

---

### Decision · Program_Chairs · 2023-10-26

Accept (Poster)